# Association between High-Sensitivity Cardiac Troponin I and Clinical Prognosis of Neurosurgical and Neurocritically Ill Patients

**DOI:** 10.3390/diagnostics12092259

**Published:** 2022-09-19

**Authors:** Jung Hwa Lee, Yun Im Lee, Joonghyun Ahn, Jeong-Am Ryu

**Affiliations:** 1Department of Neurology, Ewha Women’s University Hospital, Ewha Women’s University College of Medicine, Seoul 07985, Korea; 2Department of Critical Care Medicine, Ewha Women’s University Hospital, Ewha Women’s University College of Medicine, Seoul 07985, Korea; 3Department of Internal Medicine, National Cancer Center, Goyang 10408, Korea; 4Statistic and Data Center, Clinical Research Institute, Samsung Medical Center, Seoul 06351, Korea; 5Department of Critical Care Medicine, Samsung Medical Center, Sungkyunkwan University School of Medicine, Seoul 06351, Korea; 6Department of Neurosurgery, Samsung Medical Center, Sungkyunkwan University School of Medicine, Seoul 06351, Korea

**Keywords:** cardiac troponin I, prognosis, neurosurgery, intensive care unit

## Abstract

To investigate whether high-sensitivity troponin I (hs-TnI) elevation is associated with in-hospital mortality and major adverse cardiac events (MACEs) in neurosurgical and neurocritically ill patients. Among neurosurgical patients admitted to the intensive care unit (ICU) from January 2013 to December 2019, those whose serum hs-TnI levels were obtained within 7 days after ICU admission were included. Propensity score matching was used. Each patient with hs-TnI elevation was matched to a control patient. The primary endpoint was in-hospital mortality and the secondary outcome was MACEs. The hs-TnI elevation was shown in 848 (14.1%) of 6004 patients. After propensity score matching, 706 pairs of data were generated by 1:1 individual matching without replacement. In multivariable analysis of overall and propensity score-matched population, hs-TnI elevation was associated with in-hospital mortality (adjusted odds ratio (OR): 2.37, 95% confidence interval (CI): 1.68–3.33 and adjusted OR: 1.89, 95% CI: 1.28–2.81, respectively). In addition, hs-TnI elevation was associated with MACEs (adjusted OR: 2.73, 95% CI: 1.74–4.29 and adjusted OR: 2.64, 95% CI: 1.60–4.51, respectively). In this study, hs-TnI elevation was associated with in-hospital mortality and MACEs in neurosurgical and neurocritically ill patients.

## 1. Background

Perioperative myocardial injury is associated with major adverse cardiac events (MACEs) and clinical prognosis of patients with non-cardiac or non-vascular surgeries [1,2]. Many surgical patients experience MACEs during the perioperative period and the first year after surgery [1,3,4,5]. In particular, postoperative cardiac troponin elevation is important to predict prognosis of these surgical patients [1]. In addition, cardiac troponin elevation is associated with increased mortality and hospitalization in critically ill patients [6]. Regardless of the associated cardiovascular disease, cardiac troponin is a specific marker of myocardial injury and a predictor of prognosis [7,8,9]. From the Fourth Universal Definition of Myocardial Infarction, a diagnosis criteria of acute myocardial infarction includes the detection of an increase and/or decrease of a high-sensitivity troponin, with at least one value above the 99th percentile of the upper reference limit (URL) and at least one of the following: symptoms of myocardial ischemia, new ischemic changes in electrocardiogram, development of pathological Q waves on electrocardiogram, imaging evidence of loss of viable myocardium or new regional wall motion abnormality in a pattern consistent with an ischemic etiology or intracoronary thrombus detected on angiography or autopsy [10,11,12,13,14].

Most morbidity and mortality of neurosurgical patients may be due to neurosurgical or neurocritical illness, although cardiac injury might also contribute to their poor clinical prognosis [15,16,17]. Cardiac troponin elevation is also associated with prognosis of neurocritically ill patients with intracerebral hemorrhage or subarachnoid hemorrhage [15,17,18,19].

A limited number of studies have reported that clinical outcomes of neurosurgical and neurocritically ill patients are associated with high-sensitivity troponin I (hs-TnI) elevation [15,16,19]. Therefore, the objective of this study was to investigate whether hs-TnI elevation might be associated with in-hospital mortality and MACEs in patients admitted to a neurosurgical intensive care unit (ICU). In addition, we evaluated whether hs-TnI elevation per se was associated with poor prognosis when severity and factors other than hs-TnI elevation were controlled by propensity score matching.

## 2. Methods

### 2.1. Study Population

This was a retrospective, single-center, observational study. Patients who were admitted to the neurosurgical ICU in a tertiary referral hospital (Samsung Medical Center, Seoul, Korea) from January 2013 to December 2019 were eligible. This study was approved by the Institutional Review Board of Samsung Medical Center (approval number: SMC 2020-09-082). Included criteria were: (1) patients who were hospitalized in the neurosurgical ICU due to postoperative management or neurocritical illness and (2) those whose serum hs-TnI levels were obtained within seven days after ICU admission. Exclusion criteria were: (1) those with insufficient medical records, (2) those who had ‘do not resuscitate’ order, (3) those who were admitted to departments other than neurosurgery, and (4) those who were transferred to other hospitals or with unknown prognoses (Figure 1).

### 2.2. Definitions and Endpoints

In this study, baseline characteristics such as comorbidities, ICU management, and laboratory data were collected retrospectively using Clinical Data Warehouse. Our center constructed the “Clinical Data Warehouse Darwin-C” designed for investigators to search and retrieve de-identified medical records from electronic archives. It contains data pertaining to more than four million patients. Clinical and laboratory data were extracted from the Clinical Data Warehouse Darwin-C after finalizing the patient list in this study. Risk of surgery was defined according to the 2014 European Society of Cardiology/European Society of Anesthesiology (ESC/ESA) guidelines [20]. Perioperative management of patients followed institutional protocols based on current guidelines [9,20]. According to the institutional guideline, perioperative hs-TnI was measured for patients with more than moderate risk or undergoing moderate- to high-risk surgeries [9,20]. It was also measured at the discretion of attending clinicians for patients with mild risks [20]. An automated analyzer (Advia Centaur XP, Siemens Healthcare Diagnostics, Erlangen, Germany) with a highly sensitive immunoassay was used for hs-TnI measurement. The lowest limit of detection was 6 ng/L. The 99th percentile URL is 40 ng/L provided by the manufacturer. In this study, hs-TnI elevation was defined as an increase in hs-TnI above 40 ng/L within 7 days after ICU admission according to Fourth Universal Definition of Myocardial Infarction and the 99th-percentile URL provided by the manufacturer [9,11]. Acute Physiology and Chronic Health Evaluation (APACHE) II score was calculated based on the worst value recorded during the initial 24 h in the ICU admission [21,22]. If the patient was intubated, the verbal score of Glasgow Coma Scale (GCS) was estimated using eye and motor scores as reported previously [23]. MACEs were defined as non-fatal cardiac arrest, emergent coronary revascularization, acute coronary syndrome, stroke, congestive heart failure, atrial fibrillation (new onset or destabilization of pre-existing atrial fibrillation), major arrhythmia, cardiovascular death, and rehospitalization for cardiovascular reasons [1]. The primary endpoint was in-hospital mortality, and the secondary outcome was MACES.

### 2.3. Statistical Analyses

All data are presented as means ± standard deviations for continuous variables and frequencies and proportions for categorical variables. Data were compared using Student’s t-test for continuous variables and Chi-square test or Fisher’s exact test for categorical variables. Propensity score matching was used to control the selection bias and the confounding factor detected in this observational study. Each patient with hs-TnI elevation was matched to one control patient with the nearest neighbor matching within calipers determined by the propensity score. A caliper width of 0.2 of the standard deviation of the logit of the propensity score was used for the matching [24]. To determine the effectiveness of propensity score matching for controlling the differences between patients with and without hs-TnI elevation, standardized mean differences (SMDs) were calculated for each variable before and after matching. SMDs less than 10% indicated successful propensity scores matching and balancing between the two groups. To evaluate whether there was a difference in in-hospital mortality and MACEs according to the hs-TnI elevation, we performed multiple logistic regression with stepwise variable selection in the overall and matched population. In the overall population, we tried to obtain the result of correcting confounding through regression adjustment, and in the matching dataset, we performed doubly robust estimation to additionally correct the bias that might still exit after propensity score matching. The variables included in the multiple analyses were age, sex, comorbidities, cause of ICU admission, utilization of organ support modalities, including mechanical ventilators, continuous renal replacement therapy and vasopressors, ICP monitoring devices, hyperosmolar therapy, GCS, and APACHE II score on ICU admission. Cumulative mortality was calculated by Kaplan–Meier estimate and compared using a log-rank test. All tests were two-sided and *p* values less than 0.05 were considered statistically significant. All statistical analyses were performed with R Statistical Software (version 4.0.2; R Foundation for Statistical Computing, Vienna, Austria).

## 3. Results

### 3.1. Baseline Characteristics

A total of 12,743 patients were admitted to the neurosurgical ICU during the study period and 6004 patients were included in the final analysis. In the overall study population, hs-TnI elevation was shown in 848 (14.1%) patients (Figure 1). The mean age of all patients was 55.8 ± 15.6 years. There were 2698 (44.9%) male patients. Malignancy (50.5%) and hypertension (34.8) were the most common comorbidities. Elective vascular surgery (37.1%) and brain tumors (36.0%) were the most common reasons for ICU admission (Table 1). In the overall population, there were significant differences for variables of baseline characteristics between the two groups except for current smoking and the use of mannitol (Table 1). The mean value of maximum hs-TnI level was higher in the hs-TnI elevation group than in the normal hs-TnI group (4196.1 ± 31,274.7 ng/L vs. 8.7 ± 6.5 ng/L, *p* < 0.001). After propensity score matching, 706 pairs of data were generated by 1:1 individual matching without replacement. No significant imbalance was found in baseline characteristics between matched pairs (Table 1).

### 3.2. Clinical Outcomes

In the overall study population, rates of in-hospital mortality and 28-day mortality were higher in patients with hs-TnI elevation than in those without hs-TnI elevation (26.7% vs. 3.1% and 26.3% vs. 2.8%, both *p* < 0.001) (Table 1). Clinical outcomes in the propensity score-matched population were similar to those of the entire population. In the propensity score-matched population, rates of in-hospital mortality and 28-day mortality were also higher in the elevated hs-TnI group than in the normal hs-TnI group (19.3% vs. 14.4%, *p* = 0.019 and 19.0% vs. 13.2, *p* = 0.004, respectively). MACEs were more common in patients with hs-TnI elevation than in those without hs-TnI elevation in the overall population and the propensity score-matched population (6.6% vs. 1.1% and 7.9% vs. 2.4%, both *p* < 0.001) (Table 1).

In multivariable analysis of the overall and propensity score-matched population, hs-TnI elevation was associated with in-hospital mortality (adjusted odds ratio (OR): 2.37, 95% confidence interval (CI): 1.68–3.33 and adjusted OR: 1.89, 95% CI: 1.28–2.81, respectively). In addition, hs-TnI elevation was associated with MACES (adjusted OR: 2.73, 95% CI: 1.74–4.29 and adjusted OR: 2.64, 95% CI: 1.60–4.51, respectively) (Table 2).

In survival analysis, the mortality rates of patients with hs-TnI elevation were significantly higher than those of patients without hs-TnI elevation in the overall population and the propensity score-matched population (*p* < 0.001 and *p* = 0.008, respectively) (Figure 2).

## 4. Discussion

In this study, we investigated whether hs-TnI elevation was associated with mortality and MACEs in patients admitted to a neurosurgical ICU. Major findings of this study were as follows. First, elevated hs-TnI level was shown in 14.1% of neurosurgical patients in the overall population. Second, rates of in-hospital mortality and 28-day mortality were higher in patients with hs-TnI elevation than in those without hs-TnI elevation in the overall study population and the propensity score-matched population. Finally, multivariable analysis revealed that hs-TnI elevation was associated with in-hospital mortality and MACES in the overall and propensity score-matched populations.

Cardiac troponin is a regulatory protein that can lead to myocardial contraction by controlling calcium-mediated interaction with actin and myosin [6,25]. Destroyed cardiomyocytes can release cardiac troponin into the blood that can be detected using a commercially available immunoassay [6]. Postoperative myocardial injury is an independent predictor of cardiovascular complications and mortality within 30 days and 1 year in patients undergoing orthopedic or abdominal surgeries [1]. In particular, hs-TnI elevation is associated with worse cardiac outcomes after major surgeries [26]. In addition, elevated hs-TnI measurements among critically ill patients are associated with increased mortality and ICU length of stay [26,27]. 

In patients with subarachnoid hemorrhage, electrocardiographic abnormalities, including prolongation of QT interval and repolarization abnormalities, are commonly detected [15,18]. In particular, cardiac troponin elevation has been found in one-third of patients with subarachnoid hemorrhage known to be associated with increased mortality [15,19]. Cardiac troponin elevation is also associated with mortality in patients with surgically treated intracerebral hemorrhage and traumatic brain injury [15,28]. Under stressful conditions such as acute brain injury, stimulation of the hypothalamic paraventricular nucleus as the main control center of the hypothalamic–pituitary–adrenal axis can activate sympathetic output and lead to electrocardiographic abnormalities, arrhythmia, and myocardial injury [29]. In addition, activation of this axis after acute brain injury can cause a significant increase in catecholamines. The catecholamine surge hypothesis is the most widely accepted mechanism of brain–heart interaction [29]. Recent histological studies have shown that catecholamine-mediated myocardial injury may be a major pathophysiology of neurocritical illness [15,16,17,30] (Figure 3). In particular, the catecholamine surge leads to activation of cyclic adenosine monophosphate (cAMP) in myocytes. cAMP activation causes excessive Ca^++^ influx and altered actin–myosin interaction. The actin–myosin interaction in cardiac muscle can be prolonged beyond its physical integrity [31]. As a result, the catecholamine stimulation surge can lead to myofibrillar degeneration and contraction band necrosis [32]. These band necrotic lesions in sub-endocardium are also associated with malignant arrhythmias by involving the conducting tissue [33]. In addition, the catecholamine surge can lead to increased potassium outflow through delayed rectifier channels. These changes lead to shortened and excessive heterogeneity of action potential duration causing disturbance of rate and rhythm. Eventually, the catecholamine surge causes myofibril necrosis [31]. Therefore, cardiac injury could be accompanied by neurosurgical or neurocritical illness. It is known to be associated with clinical prognosis [15,16,17,28,30].

Neurosurgical patients with severe brain injury are more likely to develop cardiac injury and MACEs compared to those with benign diseases. Therefore, it is not easy to determine whether elevated hs-TnI itself is associated with a poor prognosis or neurosurgical patients with elevated hs-TnI will show poor prognosis because of their neurocritical illness. Therefore, a propensity score matching method was used to adjust for this confounder in this study. In brief, hs-TnI elevation was significantly associated with poor clinical outcomes of neurosurgical and neurocritically ill patients. Finally, cardiac injury may also be a contributing factor of poor clinical outcomes, although most of the morbidity and mortality could be arising from neurocritical illness [15,16,17].

This study has several limitations. First, this was a retrospective review of medical records and data extracted from the Clinical Data Warehouse. The nonrandomized nature of registry data might have resulted in a selection bias. Second, laboratory tests including hs-TnI levels were protocol-based for patients with perioperative neurosurgery. They were performed occasionally by non-protocol methods for neurocritically ill patients without neurosurgery. Third, the pathophysiology of acute coronary syndrome could not be determined for a few patients. Cardiac catheterization was not performed in these sick patients because intrahospital transport was impossible due to severe illness. Finally, the distribution of neurosurgical diseases differed from that of the general neurosurgical ICU and the proportion of patients with brain tumors was particularly high. 

## 5. Conclusions

In this study, hs-TnI elevation was associated with in-hospital mortality and MACEs in neurosurgical and neurocritically ill patients. Eventually, perioperative or neurocritical illness-associated cardiac injury could be associated with clinical outcomes of neurosurgical and neurocritically ill patients.

## Figures and Tables

**Figure 1 diagnostics-12-02259-f001:**
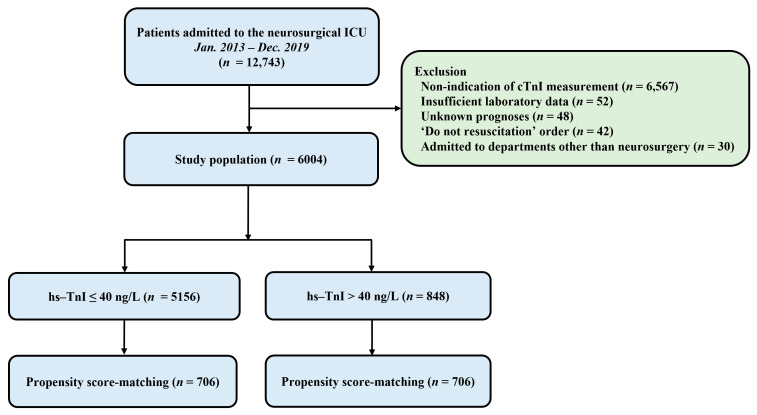
Study flow chart. ICU, intensive care unit; hs-TnI, high-sensitivity troponin I.

**Figure 2 diagnostics-12-02259-f002:**
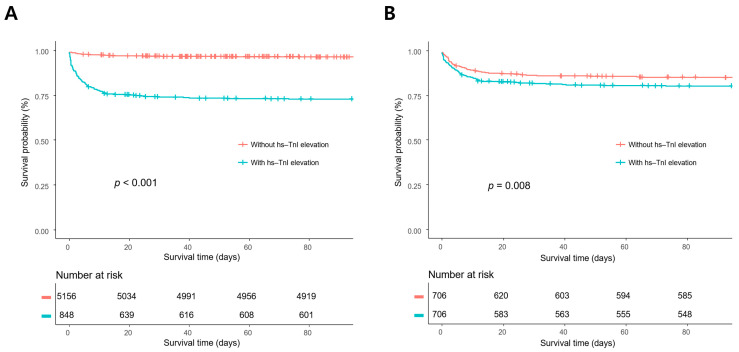
Kaplan–Meier survival analyses in the overall population (**A**) and the propensity score-matched population. (**B**) The mortality rates of the patients with high-sensitivity troponin I (hs-TnI) elevation were significantly higher compared with those without hs-TnI elevation in the overall population and the propensity score-matched population (*p* < 0.001 and *p* = 0.008, respectively).

**Figure 3 diagnostics-12-02259-f003:**
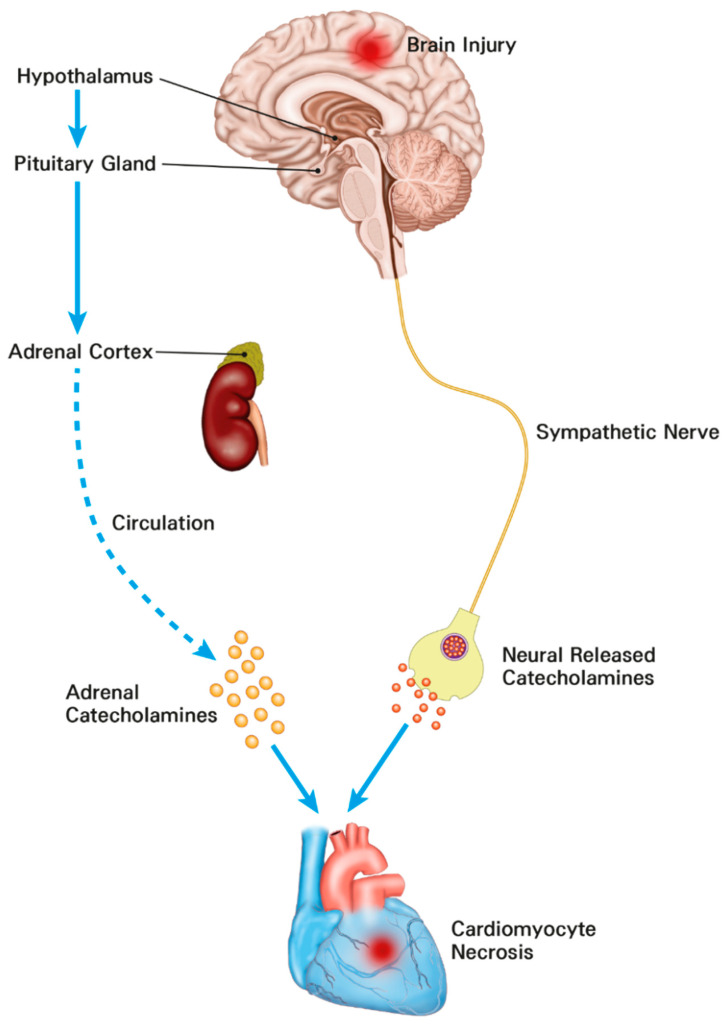
The mechanisms of cardiomyocyte damage and increased troponins in neurosurgical and neurocritically ill patients. The catecholamine surge resulting from activation of the hypothalamic–pituitary–adrenal axis and sympathetic nerve leads to myocardial damage.

**Table 1 diagnostics-12-02259-t001:** Baseline characteristics of study population.

	Overall Study Population				Propensity Score-Matched Population			
	No Elevation (*n* = 5156)	Elevation (*n* = 848)	*p* Value	SMD	No Elevation (*n* = 706)	Elevation (*n* = 706)	*p* Value	SMD
Patient demographics								
Age (year)	54.7 ± 15.3	62.1 ± 15.9	<0.001	0.471	62.0 ± 14.6	61.8 ± 16.0	0.770	0.016
Sex, male	2266 (43.9)	432 (50.9)	<0.001	0.14	349 (49.4)	366 (51.8)	0.394	0.048
Comorbidities								
Malignancy	2657 (51.5)	373 (44.0)	<0.001	0.152	350 (49.6)	344 (48.7)	0.790	0.017
Hypertension	1678 (32.5)	412 (48.6)	<0.001	0.331	341 (48.3)	347 (49.2)	0.790	0.017
Diabetes mellitus	567 (11.0)	162 (19.1)	<0.001	0.228	111 (15.7)	141 (20.0)	0.044	0.111
Chronic kidney disease	154 (3.0)	95 (11.2)	<0.001	0.324	61 (8.6)	69 (9.8)	0.519	0.039
Cardiovascular disease	92 (1.8)	88 (10.4)	<0.001	0.366	57 (8.1)	65 (9.2)	0.507	0.040
Chronic liver disease	99 (1.9)	31 (3.7)	0.002	0.106	24 (3.4)	25 (3.5)	0.999	0.008
Behavioral risk factors								
Current alcohol consumption	1257 (24.4)	172 (20.3)	0.011	0.098	151 (21.4)	147 (20.8)	0.845	0.014
Current smoking	577 (11.2)	90 (10.6)	0.662	0.019	83 (11.8)	77 (10.9)	0.675	0.027
Cause of ICU admission			<0.001	1.132			0.907	0.098
Brain tumor	1961 (38.0)	200 (23.6)			200 (28.3)	196 (27.8)		
Elective vascular surgery	2137 (41.4)	93 (11.0)			74 (10.5)	93 (13.2)		
Intracerebral hemorrhage	234 (4.5)	160 (18.9)			138 (19.5)	129 (18.3)		
Traumatic brain injury	221 (4.3)	155 (18.3)			122 (17.3)	116 (16.4)		
Subarachnoid hemorrhage	202 (3.9)	144 (17.0)			100 (14.2)	100 (14.2)		
Spinal surgery	213 (4.1)	29 (3.4)			25 (3.5)	28 (4.0)		
Central nervous system infection	41 (0.8)	10 (1.2)			7 (1.0)	8 (1.1)		
Cerebral infarction	29 (0.6)	18 (2.1)			18 (2.5)	14 (2.0)		
Others	118 (2.3)	39 (4.6)			22 (3.1)	22 (3.1)		
APACHE II score on ICU admission	3.2 ± 4.3	7.54 ± 7.83	<0.001	0.691	6.0 ± 6.3	6.41 ±7.04	0.254	0.061
Glasgow coma scale on ICU admission	14.6 ± 1.5	12.2 ± 4.2	<0.001	0.776	13.2 ± 3.3	13.1 ± 3.4	0.706	0.020
ICU management								
Use of vasopressors	112 (2.2)	103 (12.1)	<0.001	0.394	68 (9.6)	73 (10.3)	0.723	0.024
Mechanical ventilation	775 (15.0)	504 (59.4)	<0.001	1.034	374 (53.0)	367 (52.0)	0.749	0.020
Continuous renal replacement therapy	11 (0.2)	46 (5.4)	<0.001	0.319	10 (1.4)	13 (1.8)	0.674	0.034
ICP monitoring	376 (7.3)	170 (20.0)	<0.001	0.378	137 (19.4)	139 (19.7)	0.946	0.007
Use of mannitol ^a^	2250 (43.6)	349 (41.2)	0.189	0.05	295 (41.8)	286 (40.5)	0.665	0.026
Use of glycerin ^a^	508 (9.9)	267 (31.5)	<0.001	0.554	207 (29.3)	209 (29.6)	0.953	0.006
Clinical outcomes ^b^								
In-hospital mortality	158 (3.1)	226 (26.7)	<0.001		102 (14.4)	136 (19.3)	0.019	
28-day mortality	144 (2.8)	223 (26.3)	<0.001		93 (13.2)	134 (19.0)	0.004	
ICU mortality	87 (1.7)	163 (19.2)	<0.001		68 (9.6)	83 (11.8)	0.228	
ICU length of stay (hour)	57.0 ± 334.2	131.8 ± 200.2	<0.001		145.3 ± 827.1	132.8 ± 203.3	0.695	
Hospital length of stay (day)	22.9 ± 93.9	45.7 ± 205.1	<0.001		55.7 ± 237.7	38.6 ± 59.8	0.065	
Major adverse cardiac events ^b^	57 (1.1)	56 (6.6)	<0.001		17 (2.4)	56 (7.9)	<0.001	
New onset arrhythmia	53 (1.0)	24 (2.8)			16 (2.3)	24 (3.4)		
Heart failure	2 (0.0)	15 (1.8)			0 (0.0)	15 (2.1)		
Acute coronary syndrome	1 (0.0)	12 (1.4)			0 (0.0)	12 (1.7)		
Cardiac arrest	1 (0.0)	5 (0.6)			1 (0.1)	5 (0.7)		
Cardiovascular death	0 (0.0)	15 (1.8)			0 (0.0)	15 (2.1)		

Data are presented as numbers (%) or means ± standard deviations. ^a^ Some patients received more than one hyperosmolar agent. ^b^ Variables are not retained in propensity score matching. APACHE, Acute Physiology and Chronic Health Evaluation; ICP, intracranial pressure; ICU, intensive care unit; SMD, standardized mean difference.

**Table 2 diagnostics-12-02259-t002:** The relationship between elevated high-sensitivity troponin I (hs-TnI) and clinical outcomes of the overall and propensity score-matched population.

hs-TnI Elevation within 7 Days	^a^ Adjusted Odds Ratio (95% CI)	*p* Value
In-hospital mortality		
Overall population	2.37 (1.68–3.33)	< 0.001
Propensity score-matched population	1.89 (1.28–2.81)	0.002
Major adverse cardiac events		
Overall population	2.73 (1.74–4.29)	< 0.001
Propensity score-matched population	2.64 (1.60–4.51)	< 0.001

^a^ Adjusted for age, sex, comorbidities, cause of ICU admission, utilization of organ support modalities, use of invasive ICP monitoring device, hyperosmolar therapy, and APACHE II score on ICU admission. CI, confidence interval; APACHE, Acute Physiology and Chronic Health Evaluation; ICP, intracranial pressure; ICU, intensive care unit.

## Data Availability

Our data are available on Harvard Dataverse Network (http://dx.doi.org/10.7910/DVN/9HU70P, accessed on 9 December 2020).

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
