# Peer review of "Association between High-Sensitivity Cardiac Troponin I and Clinical Prognosis of Neurosurgical and Neurocritically Ill Patients"

_diagnostics, 2022, doi:10.3390/diagnostics12092259_

Round 1

Reviewer 1 Report

The authors studied the association of elevated levels of cardiac troponin T with major adverse cardiac events (MACE) in neurosurgical and neurocritically patients. 

Main comments:

- 1) Title and abstract: "Association between cardiac enzyme elevation and clinical prognosis of neurosurgical and neurocritically ill patients" - The authors determined the concentration of cardiac troponin I (high-sensitive method) (hs-TnI) and this should be reflected in the title and abstract. - "hs-TnI" instead of "cardiac enzyme".

- 2) Abstract and text: the authors used a highly sensitive method for determining cardiac troponin I (hs-TnI). - "hs-TnI" instead of "cTnI".

3) Keywords: "cardiac troponin I" instead of "cardiac enzyme".

- 4) Background: the authors should mention the current practical guidelines (the fourth universal definition of myocardial infarction, etc.), criteria for myocardial damage, features of highly sensitive methods for determining cardiac troponins. I recommend using literary sources:  1) J Am Coll Cardiol. 2018 Oct 30;72(18):2231-2264. doi: 10.1016/j.jacc.2018.08.1038. Epub 2018 Aug 25. PMID: 30153967; 2) Vasc Health Risk Manag. 2021 Jun 3;17:299-316. doi: 10.2147/VHRM.S300002. PMID: 34113117; PMCID: PMC8184290. 3) Eur Heart J. 2021 Apr 7;42(14):1289-1367. doi: 10.1093/eurheartj/ehaa575. 4) Adv Clin Chem. 2019;93:239-262. doi: 10.1016/bs.acc.2019.07.005. Epub 2019 Aug 27. PMID: 31655731. 5) Int J Mol Sci. 2021 Oct 10;22(20):10928. doi: 10.3390/ijms222010928. PMID: 34681585 , etc. 

5) page 3, line 93-95 - the units of measurement (ng\l , μg/L) of cardiac troponins differ. In accordance with the recommendations of the International Federation of Clinical Chemistry (IFCC), ng/l, but not μg/L, should be used as units of measurement of highly sensitive troponins.

6) Discussion: The authors should describe in more detail the possible mechanisms of increasing cardiac troponins in the serum of patients.

7) An additional figure ("mechanisms of cardiomyocyte damage and increased troponins in neurosurgical and neurocritically ill patients") would be useful.

                                                                     Best regards, reviewer, 04.09.2022

Author Response

September 14 2022

Dr. Vickie Xie, Assistant Editor

Diagnostics

Manuscript ID: diagnostics-1903719

Title: Association between cardiac enzyme elevation and clinical prognosis of neurosurgical and neurocritically ill patients

Dear Dr. Vickie Xie

Thank you very much for your letter and for the helpful comment from the reviewer. We appreciate the opportunity to resubmit our revised manuscript entitled “Association between cardiac enzyme elevation and clinical prognosis of neurosurgical and neurocritically ill patients”. As always, you and your editorial staff have again provided us with a comprehensive and prompt review. Many of the valuable and constructive points that the reviewers pointed out were well taken by all the authors. After going over the reviewer’s comments, my colleagues and I have performed additional investigation and made some revisions in hopes of improving our paper. The revised and added portions of the manuscript are stated in the “Response to Reviewers” and are underlined and highlighted in the revised manuscript for your convenience.

All authors contributed to the conception and interpretation of data, drafting of the manuscript, revising it critically for important intellectual content, and final approval of the manuscript. The whole manuscript or part of it, neither has been published and is not being considered for publication elsewhere in any language except as an abstract. None of the authors have any financial relationships with any company or any other bias or conflict of interest.

We believe that these findings have scientific and clinical impact and will be interesting and informative to your readers. We hope that, upon review, our study will be found to be meritorious of publication in the Diagnostics.

Jeong-Am Ryu, M.D., Ph.D.

Department of Critical Care Medicine and Department of Neurosurgery, Samsung Medical Center, Sungkyunkwan University School of Medicine, 81 Irwon-ro, Gangnam-gu, Seoul 06351, Republic of Korea

Tel: 82-2-3410-6399, Fax: 82-2-2148-7088

E-mail: lamyud.ryu@samsung.com

Response to Reviewers

Reviewer #1:

The authors studied the association of elevated levels of cardiac troponin T with major adverse cardiac events (MACE) in neurosurgical and neurocritically patients.

Main comments:

- 1) Title and abstract: "Association between cardiac enzyme elevation and clinical prognosis of neurosurgical and neurocritically ill patients" - The authors determined the concentration of cardiac troponin I (high-sensitive method) (hs-TnI) and this should be reflected in the title and abstract. - "hs-TnI" instead of "cardiac enzyme".

R1. We apologize for the use of inappropriate terminology. As your recommendation, we have revised “cardiac enzyme” to “hs-TnI” in the title and abstract. The title has been changed to “Association between high-sensitivity cardiac troponin I and clinical prognosis of neurosurgical and neurocritically ill patients”.

- 2) Abstract and text: the authors used a highly sensitive method for determining cardiac troponin I (hs-TnI). - "hs-TnI" instead of "cTnI".

R2. As your recommendation, we have revised “cTnI” to “hs-TnI” in abstract and text.

- 3) Keywords: "cardiac troponin I" instead of "cardiac enzyme".

R3. As your recommendation, we have revised “cardiac enzyme” to “cardiac troponin I” in keywords.

- 4) Background: the authors should mention the current practical guidelines (the fourth universal definition of myocardial infarction, etc.), criteria for myocardial damage, features of highly sensitive methods for determining cardiac troponins. I recommend using literary sources:  1) J Am Coll Cardiol. 2018 Oct 30;72(18):2231-2264. doi: 10.1016/j.jacc.2018.08.1038. Epub 2018 Aug 25. PMID: 30153967; 2) Vasc Health Risk Manag. 2021 Jun 3;17:299-316. doi: 10.2147/VHRM.S300002. PMID: 34113117; PMCID: PMC8184290. 3) Eur Heart J. 2021 Apr 7;42(14):1289-1367. doi: 10.1093/eurheartj/ehaa575. 4) Adv Clin Chem. 2019;93:239-262. doi: 10.1016/bs.acc.2019.07.005. Epub 2019 Aug 27. PMID: 31655731. 5) Int J Mol Sci. 2021 Oct 10;22(20):10928. doi: 10.3390/ijms222010928. PMID: 34681585 , etc.

R4. Thank you for thoughtful recommendation. As your recommendation, we have mentioned the current practical guidelines, criteria for myocardial damage, features of highly sensitive methods for determining cardiac troponins. We embedded the following sentences in the Methods section.

Line 58-65 in page 4: From the Fourth Universal Definition of Myocardial Infarction, a diagnosis criteria of acute myocardial infarction includes is the detection of an increase and/or decrease of a high-sensitivity troponin, with at least one value above the 99th percentile of the upper reference limit and at least one of the following; symptoms of myocardial ischemia, new ischemic changes in electrocardiogram, development of pathological Q waves on electrocardiogram, imaging evidence of loss of viable myocardium or new regional wall motion abnormality in a pattern consistent with an ischemic etiology, or intracoronary thrombus detected on angiography or autopsy (Ref. 10-14).

- 5) page 3, line 93-95 - the units of measurement (ng\l , μg/L) of cardiac troponins differ. In accordance with the recommendations of the International Federation of Clinical Chemistry (IFCC), ng/l, but not μg/L, should be used as units of measurement of highly sensitive troponins.

R5. We apologize for the use of inappropriate units. We revised unit “μg/L” to “ng/l” in the Method section.

- 6) Discussion: The authors should describe in more detail the possible mechanisms of increasing cardiac troponins in the serum of patients.

R6. We agree reviewer’s comment. As your recommendation, we added the following sentences in the Discussion section.

Line 237-245 in page 15: Especially, the catecholamine surge leads to activate of cyclic adenosine monophosphate in myocytes causing excessive Ca++ influx. The actin–myosin interaction in cardiac muscle can be prolonged beyond their physical integrity. As a result, the catecholamine stimulation surge can lead to myofibrillar degeneration and contraction band necrosis. These band necrotic lesions in sub-endocardium are also associated with malignant arrhythmias by involving the conducting tissue. In addition, the catecholamine surge can lead to increased potassium outflow through delayed rectifier channels. These changes lead to shortened and excessive heterogeneity of action potential duration causing disturbance of rate and rhythm. Eventually, the catecholamine surge causes myofibril necrosis.

- 7) An additional figure ("mechanisms of cardiomyocyte damage and increased troponins in neurosurgical and neurocritically ill patients") would be useful.

R7. As your recommendation, we added the following figure in the revised manuscript.

Figure 3. The mechanisms of cardiomyocyte damage and increased troponins in neurosurgical and neurocritically ill patients. The catecholamine surge resulting from activation of the hypothalamic-pituitary-adrenal axis and sympathetic nerve leads to myocardial damage.

We thank the reviewer for valuable comments. Addressing them fully has significantly strengthened the manuscript.

Reviewer 2 Report

 Dear Authors

 You performed a large retrospective analysis of  short term survival and MACE of patients hospitalized in the neurological and neurosurgical critical care unit from your hospital database . You compared patients with significantly elevated levels of cTnI versus patients without troponin elevation with adjustment for multiple confounders using propensity analysis.

 The finding of prognostic importance of  the increased  troponin levels has already been reported in many previous publications , many of them were cited in your article. I agree that less publications were dedicated to intensive neurological and neurosurgical care patients.

Major Remarks

1 As your main interest is related to cardiac troponin I ask you to describe which cardiac troponin was analized in your study as I understand  it was high sensitivity troponin I  

2. Explain in introduction or methods section what was the rationale to use cTnI cut-off value above 0.06μg/for defining increased troponin level in your study – it can be unclear to the reader

3. In this study, elevation was defined as an increase in cTnI above 0.06μg/L within 7 days after admission which is well above 99% of the detection level and fulfill the definition of the characteristic increase in cTnI for myocardial infarction (fourth universal definition).

4. Interestingly this study was probably done, with data derived from  the same or similar database  as the work referenced in your text by  Park, J.; Hyeon, C.W.; Lee, S.H.; Kim, J.; Kwon, J.H.; Yang, K.; Min, J.J.; Lee, J.H.; Lee, S.M.; Yang, J.H.; et al. Mildly Elevated Cardiac Troponin below the 99th-Percentile Upper Reference Limit after Noncardiac Surgery. Korean Circ J2020, 50, 925-937 which proved that even smaller increases of cTnI resulted in worse prognosis

5. In the study cited above by Park et al  the cut off value for the 99th percentile for cTnI was 40 ng/l – for that, a justification to use a level of 0.06μg/L in your work is needed.

6. Did you look if in your data set to a similar group of pts like Park et al. with mildly  increased cTnI? This could provide you of the entire spectrum of increased levels of troponin in this special subset of pts and possibly define a cut-off value for worse outcomes.

 7 Did you try to divide your data set of patients by median value of elevated troponin and do you find a difference in survival between patients with the upper and lower half troponin I values groups ?

 Minor remarks

1.    1.  I suggest to add in the title of the your article the name of analyzed cardiac marker to better reflect the content of your article (cardiac troponin I)

2.    Use a full name of the measured marker in the title and when it appeared for the first time in the text with abbreviation in parentheses cTnI

3.    In Fig. 2 provide (add) the number at risk at a risk of patients at given time (in parentheses on the X axis –representing survival time).

4.    3 Please correct cTn to cTnI line 180

Author Response

September 14 2022

Dr. Vickie Xie, Assistant Editor

Diagnostics

Manuscript ID: diagnostics-1903719

Title: Association between cardiac enzyme elevation and clinical prognosis of neurosurgical and neurocritically ill patients

Dear Dr. Vickie Xie

Thank you very much for your letter and for the helpful comment from the reviewer. We appreciate the opportunity to resubmit our revised manuscript entitled “Association between cardiac enzyme elevation and clinical prognosis of neurosurgical and neurocritically ill patients”. As always, you and your editorial staff have again provided us with a comprehensive and prompt review. Many of the valuable and constructive points that the reviewers pointed out were well taken by all the authors. After going over the reviewer’s comments, my colleagues and I have performed additional investigation and made some revisions in hopes of improving our paper. The revised and added portions of the manuscript are stated in the “Response to Reviewers” and are underlined and highlighted in the revised manuscript for your convenience.

All authors contributed to the conception and interpretation of data, drafting of the manuscript, revising it critically for important intellectual content, and final approval of the manuscript. The whole manuscript or part of it, neither has been published and is not being considered for publication elsewhere in any language except as an abstract. None of the authors have any financial relationships with any company or any other bias or conflict of interest.

We believe that these findings have scientific and clinical impact and will be interesting and informative to your readers. We hope that, upon review, our study will be found to be meritorious of publication in the Diagnostics.

Jeong-Am Ryu, M.D., Ph.D.

Department of Critical Care Medicine and Department of Neurosurgery, Samsung Medical Center, Sungkyunkwan University School of Medicine, 81 Irwon-ro, Gangnam-gu, Seoul 06351, Republic of Korea

Tel: 82-2-3410-6399, Fax: 82-2-2148-7088

E-mail: lamyud.ryu@samsung.com

Response to Reviewers

Reviewer #2:

You performed a large retrospective analysis of short term survival and MACE of patients hospitalized in the neurological and neurosurgical critical care unit from your hospital database. You compared patients with significantly elevated levels of cTnI versus patients without troponin elevation with adjustment for multiple confounders using propensity analysis.

 The finding of prognostic importance of the increased troponin levels has already been reported in many previous publications, many of them were cited in your article. I agree that less publications were dedicated to intensive neurological and neurosurgical care patients.

  1. Thank you for thoughtful recommendation. As your comments, elevated cardiac troponin is associated with prognosis in critically ill patients, and we knew this relationship has been reported in several publications. Neurocritically ill patients may also have elevated cardiac troponin level due to various reasons. Especially, catecholamine surge from elevated ICP may lead to cardiac injury and represent elevated cardiac troponin. However, there were limited reports about cardiac troponin elevation in neurosurgical patients and neurocritically ill patients. In this study, we investigated relationship between cardiac troponin elevation and in-hospital mortality in neurocritially ill patients.

Major Remarks

  1. As your main interest is related to cardiac troponin I ask you to describe which cardiac troponin was analized in your study as I understand it was high sensitivity troponin I

R-M1. We apologize for the use of inappropriate terminology. In this study, an automated analyzer (Advia Centaur XP; Siemens Healthcare Diagnostics, Erlangen, Germany) with a highly sensitive immunoassay was used for cardiac troponin I measurement. We have revised “cardiac enzyme”, “cTnI” to “high-sensitivity troponin I (hs-TnI)” in the revised manuscript.

  1. Explain in introduction or methods section what was the rationale to use cTnI cut-off value above 0.06μg/for defining increased troponin level in your study – it can be unclear to the reader
  2. In this study, elevation was defined as an increase in cTnI above 0.06μg/L within 7 days after admission which is well above 99% of the detection level and fulfill the definition of the characteristic increase in cTnI for myocardial infarction (fourth universal definition).
  3. Interestingly this study was probably done, with data derived from the same or similar database as the work referenced in your text by Park, J.; Hyeon, C.W.; Lee, S.H.; Kim, J.; Kwon, J.H.; Yang, K.; Min, J.J.; Lee, J.H.; Lee, S.M.; Yang, J.H.; et al. Mildly Elevated Cardiac Troponin below the 99th-Percentile Upper Reference Limit after Noncardiac Surgery. Korean Circ J2020, 50, 925-937 which proved that even smaller increases of cTnI resulted in worse prognosis
  4. In the study cited above by Park et al the cut off value for the 99th percentile for cTnI was 40 ng/l – for that, a justification to use a level of 0.06μg/L in your work is needed.

R-M2–5. Thank you for your thoughtful comments. In this study, hs-TnI elevation was defined as an increase in hs-TnI above 0.06 μg/L within 7 days after ICU admission. There were several cut-off values of hs-TnI in previous studies about relationship between cardiac troponin elevation and clinical outcomes. In this study, a cTnI values above 0.06 μg/L were considered abnormal values based on the work of Dr. Oscarsson (Ref 2). In addition, in this study, hs-TnI showed the best cut-off value at 0.063 μg/L obtained by ROC curve analysis for prediction of in-hospital mortality; cut-off value was detailed in following R-M6. As your comments, Fourth Universal Definition of Myocardial Infarction defined elevated cTn value as above the 99th percentile upper reference limit (URL) and this elevation is defined as myocardial injury. In addition, same as Park’s study (Ref 14), an automated analyzer (Advia Centaur XP; Siemens Healthcare Diagnostics, Erlangen, Germany) with a highly sensitive immunoassay was used for hs-TnI measurement in this study. The 99th-percentile URL was 40 ng/L, as provided by the manufacturer. We fully accepted your recommendation. As your word to the wise, it is more reasonable to change the cut-off value according to the Fourth Universal Definition and the information provided by the manufacturer. Therefore, we revised cut-off value 60 ng/L to 40 ng/L. Then, all statistical analyzes including propensity score matching were performed again. After re-analysis, hs-TnI elevation was also associated with in-hospital mortality and MACEs in neurosurgical and neurocritically ill patients. However, since best cut-off value was close to 60 ng/L, there were more clear difference in primary and secondary outcomes in population with over 60 ng/L defined as elevation group than over 40 ng/L.

We added the following sentence in the Method section.

Line 109-113 in page 6: The 99th-percentile upper reference limit (URL) is 40 ng/L provided by the manufacturer. In this study, hs-TnI elevation was defined as an increase in hs-TnI above 40 ng/L within 7 days after ICU admission according to Fourth Universal Definition of Myocardial Infarction and the 99th-percentile URL provided by the manufacturer (Ref. 9, 21).

Ref 2: Oscarsson, A.; Fredrikson, M.; Sorliden, M.; Anskar, S.; Gupta, A.; Swahn, E.; Eintrei, C. Predictors of cardiac events in high-risk patients undergoing emergency surgery. Acta Anaesthesiol Scand 2009, 53, 986-994.

Ref 9: Park, J.; Hyeon, C.W.; Lee, S.H.; Kim, J.; Kwon, J.H.; Yang, K.; Min, J.J.; Lee, J.H.; Lee, S.M.; Yang, J.H.; et al. Mildly Elevated Cardiac Troponin below the 99th-Percentile Upper Reference Limit after Noncardiac Surgery. Korean Circ J 2020, 50, 925-937.

Ref 21. Thygesen, K.; Alpert, J.S.; Jaffe, A.S.; Chaitman, B.R.; Bax, J.J.; Morrow, D.A.; White, H.D.; Group, E.S.D. Fourth universal definition of myocardial infarction (2018). European Heart Journal 2018, 40, 237-269.

In addition, we have revised the Results section, Table 1&2, and Figure 1&2 according to results of re-analysis in revised manuscript.

  1. Did you look if in your data set to a similar group of pts like Park et al. with mildly increased cTnI? This could provide you of the entire spectrum of increased levels of troponin in this special subset of pts and possibly define a cut-off value for worse outcomes.

R-M6. In this study, the patients with elevated cardiac troponin were included those with mild to severely elevated hs-TnI. However, median value of hs-TnI was 6 ng/L (interquartile range 6 – 14). As a result, major subjects had normal to mildly elevated hs-TnI and those were similar to the subjects of Park’s study. However, there were difference between two studies. Although Park’s research involved non-cardiac surgery, whereas this study involved neurosurgical patients and neurocritical ill patients (both those who have had surgery and those who have not).  

Following box plot shows difference of hs-TnI between survivors and non-survivors (in-hospital mortality). The level of hs-TnI was log−transformed to reduce skewness. In this study, hs-TnI was higher in non-survivors than in survivors (mean ± SD: 4,765.7 ± 33,064.5 ng/L vs. 315.5 ± 8,598.0 ng/L, p < 0.001; median with IQR: 75.5 [16.8 – 889.0] ng/L vs. 6.0 [6.0 – 11.0] ng/L, p < 0.001)

In ROC curve analysis for prediction of in-hospital mortality, hs-TnI showed relatively excellent predictive performance (C-statistic: 0.826, 95% CI: 0.803 – 0.849). In this study, hs-TnI showed the best cut-off value at 0.063 μg/L obtained by ROC curve and Youden index (sensitivity 77.6%, specificity 78.5%).

  1. Did you try to divide your data set of patients by median value of elevated troponin and do you find a difference in survival between patients with the upper and lower half troponin I values groups?

R-M7. In this study, median value of hs-TnI was 6 ng/L (interquartile range 6 – 14). Based on the median value, we divided the subjects into the upper half and lower half and re-analyzed. We re-analyzed according to 6 ng/L of cut-off and add the following table: we defined hs-TnI elevation as over 6 ng/L. As a result, although it is less predictive than before, it shows a similar trend.

Overall study population

Propensity score-matched population

Lower half

(n = 3,851)

Upper half (n = 2,153)

p value

Lower half (n = 1,434)

Upper half (n = 1,434)

p value

Clinical outcomes

In-hospital mortality

63 (1.6)

321 (14.9)

<0.001

56 (3.9)

88 (6.1)

0.008

28-day mortality

57 (1.5)

310 (14.4)

<0.001

51 (3.6)

83 (5.8)

0.006

ICU mortality

33 (0.9)

217 (10.1)

<0.001

31 (2.2)

35 (2.4)

0.709

ICU length of stay (hour)

46.6 ± 361.5

105.0 ± 221.8

<0.001

71.8 ± 580.7

85.9 ± 225.5

0.393

Hospital length of stay (day)

16.8 ± 39.5

42.7 ± 186.1

<0.001

23.4 ± 60.9

31.3 ± 85.3

0.004

Major adverse cardiac events

21 (0.5)

92 (4.3)

<0.001

11 (0.8)

89 (6.2)

<0.001

New onset arrhythmia                                  

20 (0.5)

57 (2.6)

10 (0.7)

56 (3.9)

Heart failure                                    

1 (0)

12 (0.6)

1 (0.1)

12 (0.8)

Acute coronary syndrome                                           

0 (0)

17 (0.8)

0 (0)

15 (1.0)

Cardiac arrest                                 

0 (0)

6 (0.3)

0 (0)

6 (0.4)

Cardiovascular death

0 (0)

15 (0.7)

0 (0)

8 (0.6)

Minor remarks

  1. I suggest to add in the title of the your article the name of analyzed cardiac marker to better reflect the content of your article (cardiac troponin I)

R-m1. We agree with reviewer’s comment. As the reviewer’s recommendation, the title was revised to “Association between high-sensitivity cardiac troponin I and clinical prognosis of neurosurgical and neurocritically ill patients”

  1. Use a full name of the measured marker in the title and when it appeared for the first time in the text with abbreviation in parentheses cTnI

R-m2. As the reviewer’s recommendation, we revised manuscript.

  1. In Fig. 2 provide (add) the number at risk at a risk of patients at given time (in parentheses on the X axis –representing survival time).

R-m3. As the reviewer’s recommendation, we added number at risk below the survival graph. We revised figure 2 into following figure.

  1. 3 Please correct cTn to cTnI line 180

R-m4. We apologize for the use of inappropriate terminology. We revised the abbreviation of the line.

We thank the reviewer for valuable comments. Addressing them fully has significantly strengthened the manuscript.

Round 2

Reviewer 1 Report

Dear Authors,

Thank You so much for editing and Your comments.

- The authors have improved the manuscript and now it is suitable for publication. 

                                                                                  Best Regards, 16/09/2022